# Low-THz Vibrations of Biological Membranes

**DOI:** 10.3390/membranes13020139

**Published:** 2023-01-21

**Authors:** Chloe Luyet, Paolo Elvati, Jordan Vinh, Angela Violi

**Affiliations:** 1Chemical Engineering, University of Michigan, Ann Arbor, MI 48109-2125, USA; 2Mechanical Engineering, University of Michigan, Ann Arbor, MI 48109-2125, USA; 3Biomedical Engineering, University of Michigan, Ann Arbor, MI 48109-2125, USA; 4Electrical Engineering and Computer Science, University of Michigan, Ann Arbor, MI 48109-2125, USA

**Keywords:** molecular dynamics, membranes, mechanical vibration, bacterial identification, *Staphylococcus aureus*

## Abstract

A growing body of work has linked key biological activities to the mechanical properties of cellular membranes, and as a means of identification. Here, we present a computational approach to simulate and compare the vibrational spectra in the low-THz region for mammalian and bacterial membranes, investigating the effect of membrane asymmetry and composition, as well as the conserved frequencies of a specific cell. We find that asymmetry does not impact the vibrational spectra, and the impact of sterols depends on the mobility of the components of the membrane. We demonstrate that vibrational spectra can be used to distinguish between membranes and, therefore, could be used in identification of different organisms. The method presented, here, can be immediately extended to other biological structures (e.g., amyloid fibers, polysaccharides, and protein-ligand structures) in order to fingerprint and understand vibrations of numerous biologically-relevant nanoscale structures.

## 1. Introduction

The internal motions of biological membranes have increasingly been the focus of biological research, as they provide a connection between membrane composition and many biological processes [1]. For example, membranes’ vibrations and density fluctuations have been linked to the transport of small molecules across membranes [2,3,4,5]. These processes and the membrane mechanical properties are influenced not only by the presence of transmembrane proteins and membrane composition [6,7,8,9,10], but also emerging molecular structures and their distributions play a critical role. For example, lipid asymmetry across bacterial membranes has been linked to varying susceptibility to antibiotics [11,12,13], and lipid rafts—the non-homogeneous distribution of lipids into localized regions—play a role in several biological processes [1,7,8,9].

First proposed in 1911 [14], the idea that mechanical vibrations are an identifying feature of various compounds has been extensively studied in the past century [15,16,17,18,19,20,21]. Likewise, there exists a clear link between the mechanical properties of a membrane and its properties, and in turn its functionality, which has motivated research investigating the relationship between membrane vibration and cellular activity. Recently, this line of thinking has been used to show that vibrations can be used as a means of distinguishing among microorganisms [22,23], and to study the interactions between membranes and anchored or adjacent external structures, like in bacterial biofilms [24,25]. Moreover, an increasing number of studies have identified modes of membrane-adjacent structures. For example, functional amyloid fibers, proteinaceous fibers that grow in biofilm and anchor to the bacterial membranes, have been suggested to mechanically vibrate and deliver a damped vibrational signal to an adjacent bacterial cell [24,25,26]. Electromagnetic signals on the order of kHz of bacterial DNA that match DNA extracted from Alzheimer’s and other amyloid-induced diseased patients [27] suggest that bacterial infections are present in such illnesses. THz vibrations have also been observed in protein-ligand binding [28] and other biological polymers [29,30], suggesting that protein-ligand interactions trigger unique changes in vibration that can be used in detection and diagnoses.

Despite these promising results, work in this direction has been hindered by several factors. Experimentally, membranes’ mechanical and structural characterization, as well as cellular identification, have been expensive and time-consuming [1,7,8,9,21,22,23]. Indeed, many of the early works that discussed the use of mechanical vibrations as an identification tool speculated that computation would eventually dominate the field [16,18,19]. Nonetheless, while computationally probing the vibrational modes of biological structures, like membranes, is simpler, it is computationally demanding, which has led to less accurate approaches (e.g., coarse-graining, continuous models) and assumptions (e.g., membrane composition and structure) [31,32,33,34,35,36,37,38] of limited usefulness or reproducibility. Finally, even when data is available, an unbiased method for the identification and comparison of the vibrational spectra has long been a complex challenge [18,20,39,40,41,42].

To fill this gap, we propose an approach that combines atomistic molecular dynamics simulations, to gather information about the low-THz vibration of disparate membranes, with signal processing, to identify and compare their vibrational spectra. Using this approach, we discuss the effect of membrane asymmetry and lipid composition (with and without sterols) on the vibrations, as well as some hidden pitfalls that are potentially introduced by the use of atomistic simulations. Moreover, by employing a nonparametric test, our comparisons allow us to test the variability among samples obtained of the same system, and quantify spectral uncertainty.

## 2. Materials and Methods

The approach used in this work is detailed below and illustrated in Figure 1 and Figure 2. All abbreviations are listed at the end of the manuscript. Briefly, we used Molecular Dynamics (MD) to simulate different cellular membranes, and used the time evolution of the atomic positions to compute the infrared absorption in the THz region. The resulting spectra were then filtered to help with peak identification and compared to obtain a simple measure of the difference between spectra.

### 2.1. Systems

In this work, instead of taking the standard approach of exploring the effect of each possible parameter (e.g., the concentration of each possible lipid), we focused on three complex membranes using realistic compositions, as discussed in the following. We chose this approach because looking into the effect of all the possible parameters of a membrane is a generally daunting task, given the number of possible lipids and their concentrations (e.g., *B. Subtilis* has at least 127 different lipids), and their often nonlinear relations. The latter is a critical consideration, as it can make the effort of decomposing the problem in simpler tasks very challenging; the addition of a single type of lipid can markedly change certain observed properties, like the spectra in the range of interest here. We show such an example at the end of this manuscript, when discussing the effect of sterols and LPG.

With this in mind, instead of trying to create a general model, we aimed to (1) determine if we could detect any difference between real compositions and (2) discuss the problems simulations would encounter in sampling more realistic and therefore complicated systems. We studied the plasma membranes of three types of cells, two of bacterial (*Staphylococcus aureus* and *Bacillus subtilis*) and one of mammalian (rat liver plasma [43]) origin. We chose *S. aureus* because of its high pathogenicity and prevalence in hospital-acquired infections [44,45], *B. Subtilis* thanks to its ubiquity and innocuousness in healthy individuals [46], and rat liver cell membrane as an example of mammalian plasma membranes. For *S. aureus*, we considered two asymmetric membranes with different composition (S476${}_{33}^{A}$, S476${}_{51}^{A}$), observed at two different values of pH (i.e., 5.5 and 7.4), as well as a symmetric membrane (S476${}_{33}$) as close as possible to S476${}_{33}^{A}$, to study the effect of lipid distribution between leaflets [11,31,47,48,49,50,51,52]. For *B. Subtilis* and rat liver cells, however, we could only find information about the total composition of the plasma membranes and therefore, we simulated homogenous bilayers. These five systems are illustrated in Figure 1. Additionally, we simulated four membranes, derived from S476${}_{33}^{A}$ and S476${}_{51}^{A}$ by adding to each one of them 1.3% molar [53] of either ergosterol or cholesterol. For all the membranes, 3 to 9 independent replicas were generated and simulated.

Except when noted otherwise, the *S. aureus* membranes consist of a periodic bilayer of 15 nm × 15 nm in size (approximately 700 lipids, exact number depends on composition), the *B. Subtilis* membrane 16 nm × 16 nm in size (840 lipids), and the rat liver membrane, 12 nm × 12 nm in size (600 lipids). These sizes were chosen based on the estimated lowest frequency mode ($f_{min}$) that could be observed,
(1)$$f_{min}=\frac{s_{membrane}}{L_{max}}$$
where $s_{membrane}$ is the speed of sound in the membrane and $L_{max}$ the longest distance between two points on the x-y dimension (membrane plan) of the periodic box. Since $s_{membrane}$ is hard to estimate accurately, we conservatively used the speed of sound of water (1550 m s^−1^), which is greater than alcohols and alkenes with long aliphatic chains (1150–1250 m s^−1^), which places an upper limit to $f_{min}$ of approximately 0.1 THz.

### 2.2. Molecular Dynamics Simulations

All the systems were prepared using the Membrane Builder in CHARMM GUI [54] and, then, post-processed when cropped systems were needed. Nanoscale Molecular Dynamics [55] software was used to perform Molecular Dynamics simulations, a time step of 2 fs was employed to integrate the equations of motion, while hydrogen atoms were kept rigid via the SHAKE algorithm. Membranes were fully solvated in a 0.15 m NaCl solution using TIP3P for water [56] and CHARMM, version 36 [57], to model atomic interactions. Non-bonded short-range interactions smoothly approached 0 using an X-PLOR switching function between 1 nm and 1.2 nm, in conjunction with the particle mesh Ewald algorithm, to evaluate long-range Coulomb forces.

The systems were equilibrated using constrained canonical simulations, followed by isothermal-isobaric ensemble simulations with vanishing restraint, as per CHARMM GUI protocol. This preliminary equilibration was followed by 100 ns simulation in an NPsT ensemble, an isothermal-isobaric ensemble where changes in dimensions in the direction of the membrane’s plane are coupled. Starting from the equilibrated systems, spectra were computed from 50 ns microcanonical ensemble simulations, while Langmuir isotherms were computed from canonical simulations 20 ns long. In all cases, temperature was kept constant at 310 K by using a Langevin thermostat with a period of 1 ps, and pressure was imposed by using a Nosé-Hoover Langevin piston method, with a period of 200 fs and 50 fs decay. Processing and preparation of trajectories and structures were performed with the help of Visual Molecular Dynamics (VMD) software [58], as well as the MDAnalysis and Scipy Python libraries [59,60,61].

### 2.3. Spectra

Absorption cross-section $\alpha_{cs}$ as a function of the frequency ω were calculated from trajectories, using the relation:(2)$$\alpha_{cs}\left(\omega \right)=\frac{4\pi^{2}}{3\hbar c\epsilon n(\omega )}\omega \left(1-e^{-\beta \hbar \omega }\right)Q\left(\omega \right)I_{cl}\left(\omega \right)$$
where *ℏ* is the reduced Plank constant, *c* is the speed of light in vacuum, ϵ the vacuum permittivity, n(ω) the refractive index of the solution, $\beta (=1/k_{B}T)$ is the inverse of the Boltzmann constant times the temperature, Q(ω) is the harmonic quantum correction [62],
(3)$$Q_{harm}\left(\omega \right)=\frac{\beta \hbar \omega }{1-e^{-\beta \hbar \omega }}$$
and $I_{cl}\left(\omega \right)$ is the classical spectral density,
(4)$$I_{cl}\left(\omega \right)=\frac{1}{2\pi }\int_{-\infty }^{\infty }dt\;e^{-i\omega t}\langle M\left(0\right)M\left(t\right)\rangle$$
where angular brackets indicate ensemble averaging and M is the membrane’s total dipole moment. For practical reason, we assumed n(ω) = 1 as, in this range, the correction is almost linear [63]. Finally, numerical noise was reduced by using Blackman windowing when computing the Fourier transform and by applying a Savitsky-Golay filter (polynomial order of 6 over 21 windows, Appendix A) on $\alpha_{cs}\left(\omega \right)$.

Since different membranes are represented by periodic systems of different size, to compare different spectra and obtain size-independent quantities, we divided each spectra by the average value of the signal in our interval
(5)$$N=\int d\omega \;\alpha_{cs}\left(\omega \right)$$
and used $\alpha \left(\omega \right)=\alpha_{cs}\left(\omega \right)/N$ in all the comparisons.

### 2.4. Signal Analysis

Peaks were identified from the spectra by tuning the (1) prominence (vertical distance between the peak and its lowest contour line) and the (2) minimum distance between peaks. Prominence, alone, is not used as a peak-finding metric because doing so would only select the highest intensity peaks, which are located on the right side of the spectra in this case. Instead, after setting the minimum peak-to-peak distance ( 125 GHz for the rat liver cells, and 142 GHz for bacterial membranes), we computed the prominence distribution for different prominence thresholds. A demonstration of how prominence and distance between peaks are tuned is available in the Appendix A. We then identified the range of values that were closer to the average number of peaks (computer over all the prominence value) and selected the highest value. For each membrane, we compute the average frequency and intensity of each peak using the peaks computed over replicas (3 replicas for S476${}_{51}^{A}$, S476${}_{33}$, *B. Subtilis*, and rat liver; 4 replicas for S476${}_{33}^{A}$, see a more detailed discussion in the Results section).

To quantify the similarity between two spectra, we used a two sample Kolmogorov-Smirnov (KS) test [64]. As replicas of the same membrane are (or should be, see the discussion in the results) statistically equivalent, variation between replicas of the same system can be assumed to originate from the computational method. As such, when comparing different membranes, we used the mean KS (and the corresponding standard error) of the replicas of a membrane as a baseline for the comparison with other systems. Of note, all the comparisons were performed by computing the KS value for all the possible pairs of replicas of one system with another, and not by comparing the average spectra.

### 2.5. Root-Mean-Squared Fluctuations (RMSF)

RMSF, that is the mean deviation from the average position of an atom, was calculated from the microcanonical simulations, using the Python MDAnalysis module [60,61]. Due to a systematic numerical bias in the RMSF value for molecules that are close to the periodic boundaries, all the contributions from these molecules were removed. To allow a meaningful comparison with other membranes, we only computed the RMSF of the atoms that are not part of sterol molecules.

## 3. Results

Before analyzing the differences among replicas of different membranes, we tested the force field and relaxation protocol. As atomistic molecular dynamics has been extensively proven in the literature to be suitable to model bilayer dynamics, we performed a minimal validation by computing the Langmuir isotherms in the between −4.2 MPa and 1.5 MPa for S476${}_{33}^{A}$, S476${}_{51}^{A}$, and S476${}_{33}$. The comparison with experimental data (see Appendix A) shows that the difference of our estimates is well below the uncertainty (standard deviation).

As a second step, we tested the potential bias introduced by using periodic boundary conditions (i.e., size effect). To this end, we compared the spectra of membranes with identical composition (S476${}_{33}^{A}$) but having four different periodic system sizes (15 nm × 15 nm, 4 nm × 15 nm, 3 nm × 15 nm, and 3 nm × 12 nm). The results show (see Appendix A) that peak location is unaffected by the size of the bilayer patch, but the normalized intensity is marginally weaker for the smallest membranes. In the following, however, we will only use square periodic patches (15 nm × 15 nm for *S. aureus* systems, 16 nm × 16 nm for *B. Subtilis*, and 12 nm × 12 nm for rat liver, see Section 2), to avoid introducing any anisotropy in the systems.

### 3.1. Membrane Asymmetry

The asymmetry in the membrane composition is suggested to play a key role in many cellular processes. At the same time, accurate information about the distribution of species between leaflets is generally scarce, due to the difficulty of experimental measurements, as well as the dynamic nature of the cellular membrane make-up. As a larger number of average compositions are available in the literature, we compared the differences in the vibrational spectra between symmetric and asymmetric *S. aureus* systems (S476${}_{33}^{A}$ and S476${}_{33}$). The initial comparison (see Figure 3), while showing a statistical equivalence between the peaks of the two systems, was surprisingly affected by large uncertainty, despite the number of replicas used for each system (9 for S476${}_{33}$ and 6 for S476${}_{33}^{A}$).

To investigate the rationale behind this observation, we looked into the similarity among replicas of a given membrane. To make sense of all these comparisons, we built an undirected weighted graph (see Appendix A), where each node is a replica and the weight of each edge is equal to 1-KS (i.e., similar spectra are connected by an edge with higher value). Different types of clustering analyses can be performed to obtain such a graph, but the consistent result is that the replicas are not separated, as expected, in two groups based on their leaflet symmetry, but rather in three groups, where both types of systems are somewhat mixed. Given the microcanonical nature of the simulations used to generate the spectra, the reason for this clustering is a dependence on the initial conditions, likely resulting in some violation of the ergodic hypothesis. To narrow down the source of this difference, we considered the effects of the membrane thickness (see Appendix A), periodic system size, and surface tension (see Appendix A) on the spectra, but we found no strong correlation in all cases. Thus, we hypothesize that the differences are related to slightly different stability in the vibrational modes that are sampled, due to slightly different initial velocity distribution. This hypothesis is corroborated by the analysis of the peaks, which shows that, with the notable exception of the lowest frequency (∼0.2 THz), the peaks display variability in intensity and location between groups (see Appendix A).

This analysis leads to three main conclusions. First, it shows that the leaflet symmetry, for equivalent total composition, does not have a statistically significant effect on the spectra, whether we consider KS statistic of the average of all replicas or we restrict ourselves to one of the replica clusters, like in Figure 3. Of note, the membrane symmetry can still affect other processes, as well as other mechanical properties. Second, despite the differences and the complexity in comparing the spectra of each replica, the lowest frequency peak (between 0.17–0.21 THz) is conserved in all *S. aureus* systems and replicas. Finally, the spectra obtained from simulations should be carefully tested for internal consistently. Even though we observed statistical variability only among replicas of two *S. aureus* systems in this work, this issue should be tested to avoid adding systematic uncertainty to the results, especially as we will show, below, for more rigid membranes.

### 3.2. Cell Type

After establishing the effect of composition and lipid distribution for *S. aureus* membrane, we compared the spectra of the plasma membrane of different cells, namely we compared *S. aureus* with one other common bacterium as well as a mammalian cell.

The results (Figure 4) show that the *S. aureus* absorption spectra in the low-THz region is rather distinct from the other two, with a characteristic peak, just below 0.2 THz. Of note, this distinction holds regardless of the *S. aureus* replicas or the symmetry chosen (see Appendix A). *B. Subtilis* and rat liver cell are also statistically distinct, although notably more similar than *S. aureus*, despite the remarkably different composition. Notably, these differences are the results of the interplay between lipids and not a simple inertial behavior due to the difference in the membranes’ density: the mass per unit area of S476${}_{33}^{A}$ (approximately 2.4 kDa nm^−2^) falls between the one for *B. Subtilis* (approximately 2.2 kDa nm^−2^) and the mammalian cell (approximately 2.6 kDa nm^−2^).

### 3.3. Sterols

Finally, we looked in the effect of the presence of sterols on the *S. aureus* membrane. Bacterial cells do not typically synthesize sterols, as the bacterial cell wall occupies the same function fulfilled by sterol-containing plasma membranes in eukaryotic cells by maintaining structural integrity and fluidity. However, cell-wall-deficient forms of *S. aureus*, called L-forms, do exist [65]. As staphylococcal L-forms lack cell walls, sterols provide a means of maintaining structural integrity and fluidity [66]. More importantly, the presence of sterols has been linked to increased resistance to antimicrobial peptides [67] and lipid raft formation, demonstrating their importance in membrane function and biological processes. For this reason, we analyzed the effect of sterols on the absorption spectra by comparing the spectra of the asymmetric *S. aureus* membranes (without sterols), with identical membranes to which we added either 1.3% cholesterol or ergosterol [53].

The presence of sterols in the membranes (see Figure 5) had a greater effect on the spectra of S476${}_{33}^{A}$ than that of S476${}_{51}^{A}$. This difference can be related to the different content of LPG between the two membranes, as higher levels of LPG decrease the membrane fluidity [12,68,69,70,71], causing a slight change in the absorption spectra. These results agree with a study in which mutant bacteria, producing less LPG, have membranes with a reduced rigidity [72]. Indeed, the presence of high LPG concentration, much like low concentration of sterols, has been shown to have a stabilizing effect on membrane fluidity [68,69]. While these effects do not compound (S476${}_{51}^{A}$), a membrane with a lower concentration of LPG (S476${}_{33}^{A}$) would be more susceptible to changes caused by sterols. It is interesting that sterols in the membrane with more LPG cause an increase in mobility, as measured by the average atomic RMSF, while sterols in the membrane with less LPG cause an increase in rigidity (Figure 5C). Finally, cholesterol typically affects the membrane mechanical properties more than ergosterol (Figure 5B), which agrees with the experimental observation that ergosterol has a smaller effect on membrane mobility than cholesterol [67].

## 4. Discussion

The unique absorption spectra of biological membranes are a promising metric for species differentiation and bio-process identification. In this work, we show how to estimate, analyze, and compare the absorption spectra of bacterial and mammalian membranes, by combining molecular dynamics simulations and signal processing techniques (i.e., peak detection and KS statistics). The analysis of *S. aureus*, *B. Subtilis*, and rat liver cells, shows that distinct peaks can be identified for different species in the low-THz region and that certain peaks, for *S. aureus* around 0.19 THz, are present even for different compositions and symmetries, making them good identifiers under a variety of conditions.

The ability to rigorously compare noisy and complex spectra, like the one studies here, opens the door to finding unexpected correlations. For example, *S. aureus* membranes (at 7.4 pH) have a lower LPG content, which we found is associated with both increased variability among replicas and higher susceptibility to changes in mobility in the presence of sterols. This change could speak to higher rigidity in membranes with more LPG, as LPG tends to maintain membrane fluidity [12,68,69,70,71]. While higher fluidity may have biological advantages, it seems inversely correlated to antibiotic resistance as, generally, an increase in resistance of membranes to antimicrobial peptides is associated with higher concentrations of LPG [68,73] and sterols [67].

The comparison of spectra of different samples of the same system also provides a way to find similarities, even for complex signals like the one presented here. Different samples can be clustered and analyzed by building fully connected graphs, where edges are weighted based on the values of a two-sample KS test. This representation, allows visualizing and finding similarities on high dimensional spaces, like the 105th-dimensional space of the comparisons between 15 *S. aureus* samples in this work. While we found that the existence of clusters among the samples affected only the most rigid membrane (see the previous discussion about LPG content), it is nevertheless allowed extracting data from samples that would have been otherwise affected by a high uncertainty.

Finally, while this work was designed around the low-THz absorption spectra of plasma membranes, it can immediately generalize to other structures (e.g., fibers present in biofilm matrix), other regions of the vibrational spectra, as well as to data obtained from experimental techniques. Our work reinforces the idea that vibrations can be used to successfully differentiate between different biological complexes and how they are affected by specific changes, which, then, can potentially be related to biological functions. This link, if present, could inform a route by which these vibrations could be manipulated, for very targeted effects [27].

## Figures and Tables

**Figure 1 membranes-13-00139-f001:**
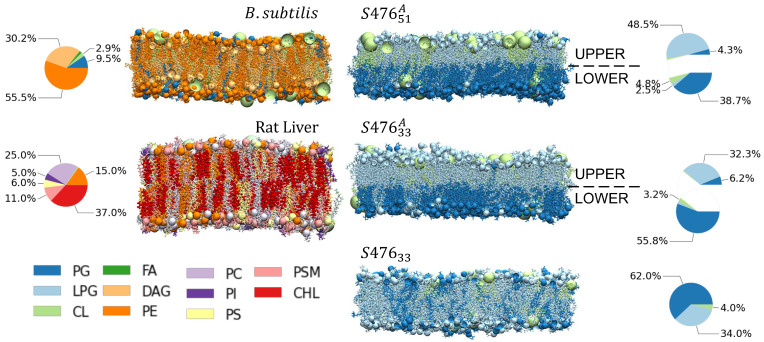
Systems and Nomenclature Summary. Composition and distribution of five types of plasma membranes modeled in this work: one for *B. Subtilis*, one for rat liver cell, and three for *S. aureus* (S476) membranes. *S. aureus* membrane are labeled according to symmetry (A for asymmetric) and percent concentration of LPG (33% or 51%). Only composition fractions greater than 2% are shown here. Only non-sterol containing membranes are shown. All abbreviations are listed at the end of the manuscript.

**Figure 2 membranes-13-00139-f002:**
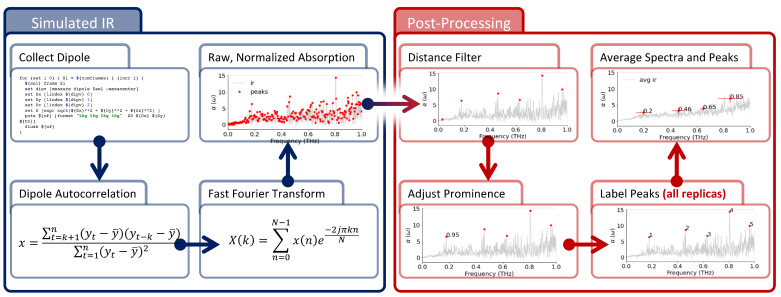
Spectra Calculation and Signal Analysis. Infrared spectra are computed from MD trajectories (blue panel) and, then, post-processed to find local maxima. Spectra are computed from MD trajectories as Fourier transform of the membrane total dipole autocorrelation; then, the spectra are normalized before peaks are identified using a distance and prominence filter (Appendix A). For more details, see Section 2.

**Figure 3 membranes-13-00139-f003:**
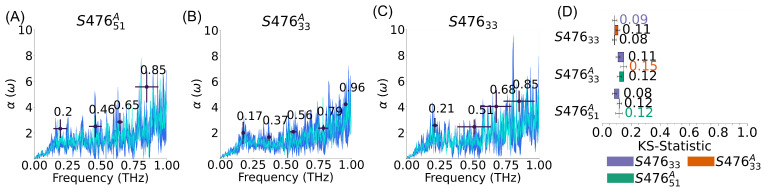
Effect of composition and distribution on *S. aureus* membrane’s spectra. (**A**–**C**) *S. aureus* membrane average spectra (cyan), standard deviation of spectra (dark blue), and peaks (black); error bars represent standard deviations. For S476${}_{33}^{A}$ and S476${}_{33}$ the spectra are computed from the largest cluster of replicas (see Appendix A). (**D**) Kolmogorov-Smirnov (KS) statistics; error bars represent the standard error of the mean. Self comparison (e.g., S476${}_{33}$/ S476${}_{33}$) indicates the average difference between replicas of the same system. KS statistic shows that this part of the absorption spectra for the three systems are not distinct.

**Figure 4 membranes-13-00139-f004:**
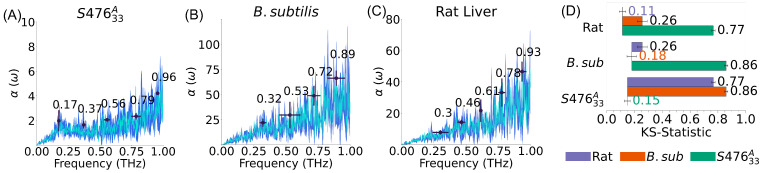
Plasma membranes absorption spectra of different species. (**A**–**C**) Average spectra are shown in cyan, standard deviation in spectra (dark blue), and peaks are labeled by black points; error bars represent standard deviations in peak location and intensity. The *S. aureus* membrane is the only one among the three that has a peak in the (0.17–0.2 THz) region. (**D**) Kolmogorov-Smirnov (KS) statistics; error bars represent the standard error of the mean. Self comparisons (e.g., Rat/Rat) indicate the average difference between replicas of the same system. KS statistic shows that *S. aureus* spectra is very distinct from the other two, even though *B. Subtilis* and Rat spectra are still statistically discernible.

**Figure 5 membranes-13-00139-f005:**
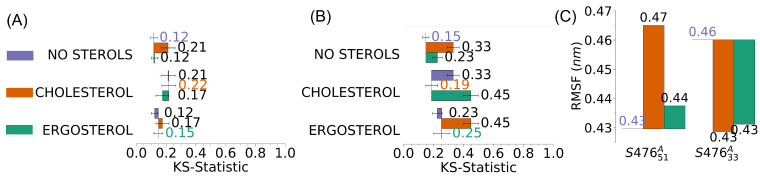
Difference in the spectra due to the presence of sterols in *S. aureus* membranes. Plots show the results of two-sample Kolmogorov-Smirnov (KS) tests for the average spectra of (**A**) S476${}_{51}^{A}$ and (**B**) S476${}_{33}^{A}$, without any additional sterol, and with 1.3% cholesterol or ergosterol. Error bars represent the standard error of the mean. (**C**) Average mobility measured as root-mean-squared fluctuations (RMSF) of the positions of non-sterol atoms in S476${}_{51}^{A}$ and S476${}_{33}^{A}$ membranes, without sterols (baseline) and with cholesterol (orange) or ergosterol (green).

## Data Availability

The data presented in this study are openly available in University of Michigan DeepBlue Documents at 10.7302/6571.

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
