# Peer review of "Low-THz Vibrations of Biological Membranes"

_membranes, 2023, doi:10.3390/membranes13020139_

Round 1

Reviewer 2 Report

The authors presented an interesting study of simulating the vibrational spectra of 5 membrane models with distinctive compositions and symmetries using molecular dynamic simulation. The auto correlation of the total dipole moment was simulated based on the trajectories of MD, and the vibrational spectra were extracted using Fourier transformation thereafter. The results suggested that S. aureus membrane, which is rich in phosphatidylglycerol (PG) and lyso-phosphatidylglycerol (LPG), has a unique vibrational mode in the low terahertz range (0.19THz) that distinguish it from B. subtilis membrane (rich in phosphatidylethanolamine) and rat liver membrane (rich in cholesterol). They also found that 1.5% sterols in the S. aureus membrane is enough to introduce significant changes in the spectra feature using Kolmogorov-Smirnov (KS) test, however the exact feature is not given by the authors. 

Overall the analysis looks sound, and the results are interesting, however, I have to admit that I got confused by several points throughout the experimental design and discussion, and would hope the authors to clarify:

1. I might have missed it, but what is the fatty acid chains used for each lipid type? I only noted the phosphate head group listed in the Figure 1 caption. As the authored noted in the study, the rigidity of the membrane basically determines the vibrational spectra, and the fatty acid group is one the most significant factors affecting membrane mobility.

2. The authors' rationale for the choice of membrane system was to cover two bacterial and one mammalian membrane. However, I found the choice of such complicated membrane system to work against one of the goals of the study, which is to distinguish different membranes based on the vibrational spectra. Why not control the lipid composition such that only phosphate group type (charge state)/fatty acid type/cholesterol is varied? These parameters are known to significantly contribute to the membrane mechanical property and polarity, which should affect the vibrational spectra. In fact, the authors partial did this by varying the including sterol in S. aureus membrane system and acknowledge that the correlation with LPG content might be associated with LPG modulating the membrane rigidity.

3. Can the author elaborate a bit more on the signal analysis section? I have hard time understand the peak identification paragraph (partly because SI is not available at the time of my review). Can you give a quantitative criterion for the tuning of prominence and distance between the peaks for the peak picking, if it's available. Given the authors argued S. aureus have a unique peak at 0.19THz, the criteria for peak picking should be stated in the main text.

There are several other minor points worth mentioning:

4. The SI is not available at the time of my review. Instead, the linked file is a reply to the reviewer by the author.

5. Figure 1b is too small.

6. In line 229 stated "as higher levels of LPG decrease the membrane fluidity", while in line 255 stated "LPG tends to maintain membrane fluidity". I thought these statements are contradictory.

Round 2

Reviewer 1 Report

I think the most important comments / changes have been addressed in the text, some of my questions came from the lack of SM, some of the concerns will be covered by the deposit. Maybe there are still  few issues open for the discussion but of course (a) we can have slightly different opinions in minor points and more importantly (b) the open discussion it doesn't halt the publication. 

Because of changing the figures location / numbers the careful proof-reading is required, for example in line 62: "Figure ??". 

Reviewer 2 Report

I appreciate the authors to provide the relevant content in the SI to clarify my confusion. I agree the SI would have answer some of my questions if it was available earlier. Unfortunately, I still don't have access to the SI. I hope the issue would be resolved later.

I still partially disagree with the authors' decision to use three drastically different membrane systems for the study. However, I also noticed the complication in the parameter space, and agree that exploring it exhaustively would be impractical. I hope the authors would add discussion of the caveats in the current experiment design and the reason why it is still chosen in the end, just like the reply to me. 

Round 3

Reviewer 2 Report

The authors have addressed the question I raised, and I think this paper should be suitable for publication.